# The Effects of Timing of a Leucine-Enriched Amino Acid Supplement on Body Composition and Physical Function in Stroke Patients: A Randomized Controlled Trial

**DOI:** 10.3390/nu12071928

**Published:** 2020-06-29

**Authors:** Takashi Ikeda, Nobuo Morotomi, Arinori Kamono, Saki Ishimoto, Ryo Miyazawa, Shogo Kometani, Rikitaro Sako, Naohisa Kaneko, Mamoru Iida, Nobuyuki Kawate

**Affiliations:** 1School of Nursing and Rehabilitation Sciences, Showa University, Tokaichiba 1865, Midori ward, Yokohama, Kanagawa 236-0001, Japan; a.kamono77@nr.showa-u.ac.jp; 2Rehabilitation Center, Showa University Fujigaoka Rehabilitation Hospital, Fujigaoka 2-1-1, Aoba ward, Yokohama, Kanagawa 236-0001, Japan; miyazawa.2223@gmail.com (R.M.); dad_20_pt@yahoo.co.jp (S.K.); sako1286@cmed.showa-u.ac.jp (R.S.); 3Research Institute for Sport and Exercise Sciences, Showa University, Fujigaoka 2-1-1, Aoba ward, Yokohama, Kanagawa 236-0001, Japan; 4Department of Rehabilitation Medicine, Showa University Fujigaoka Rehabilitation Hospital, 2-1-1, Aoba ward, Yokohama, Kanagawa 236-0001, Japan; uoratubon@gmail.com (N.M.); naohisakaneko@gmail.com (N.K.); mamo1130iida@yahoo.co.jp (M.I.); kawate@med.showa-u.ac.jp (N.K.); 5Department of Health and Welfare, Health promotion section, Yamato, Kanagawa 236-0001, Japan; saki.ishimoto@city.yamato.lg.jp

**Keywords:** branched chain amino acid, stroke, body composition, timing of supplementation

## Abstract

The combination of exercise and nutritional intervention is widely used for stroke patients, as well as frail or sarcopenic older persons. As previously shown, supplemental branched chain amino acids (BCAAs) or protein to gain muscle mass has usually been given just after exercise. This study investigated the effect of the timing of supplemental BCAAs with exercise intervention on physical function in stroke patients. The participants were randomly assigned to two groups based on the timing of supplementation: breakfast (*n* = 23) and post-exercise (*n* = 23). The supplement in the breakfast group was provided at 08:00 with breakfast, and in the post-exercise group it was provided just after the exercise session in the afternoon at 14:00–18:00. In both groups, the exercise intervention was performed with two sessions a day for two months. The main effects were observed in body fat mass (*p* = 0.02, confidence interval (CI): 13.2–17.7), leg press strength (*p* = 0.04, CI: 94.5–124.5), and Berg balance scale (*p* = 0.03, CI: 41.6–52.6), but no interaction with intake timing was observed. Although the effect of the timing of supplementation on skeletal muscle mass was similar in both groups, BCAA intake with breakfast was effective for improving physical performance and decreasing body fat mass. The results suggest that a combination of BCAA intake with breakfast and an exercise program was effective for promoting rehabilitation of post-stroke patients.

## 1. Introduction

In recent years, combinations of exercise and nutritional interventions for frail older persons or sarcopenia patients have been actively investigated [1,2,3,4,5,6,7,8]. For nutritional intervention, protein or branched chain amino acid (BCAA) supplements have been viewed with keen interest. Some systematic reviews [3,4,5,6,7] showed the efficacy of a combination of exercise and protein supplementation for improving muscle strength and physical function in frail older persons. The same [7,8] was also reported in sarcopenic persons. Furthermore, a recent systematic review [3] showed that protein supplementation alone did not improve muscle mass, muscle strength, or physical function in frail older persons. 

On the other hand, Foley et al. [9] reported that energy and protein intake were 10–20% below the required amounts of patients after strokes. Kokura et al. [10] and Nishijima et al. [11] reported that this deficiency affects the ability to improve functional independence measure (FIM) scores. Furthermore, Yoshimura et al. [12] showed that BCAA supplementation immediately after exercise led to improved FIM scores, grip strength, and skeletal muscle mass in stroke patients.

Such combined therapy has been widely used, not only for frail or sarcopenic older persons, but also for stroke patients. As previously shown, BCAA or protein supplementation for the purpose of gaining muscle mass has generally been given just after exercise. A comparison study of supplementation before and after exercise [13] showed that it was more effective post-exercise than pre-exercise. However, considering the bias of a meal including protein intake and interval time, a recent randomized study [14] reported that protein supplementation with breakfast stimulated post-prandial muscle protein synthesis and increased muscle mass in healthy older persons. In addition, Pekmez et al. [15] showed that protein intake before a meal affected the post-prandial metabolic pattern in persons with metabolic syndrome. These findings suggest the hypothesis that protein or BCAA supplementation with breakfast is more effective for gaining muscle strength and muscle mass than supplementation immediately after exercise.

This study, therefore, investigated the effect of the timing of BCAA supplementation with exercise therapy on improving skeletal muscle mass, muscle strength, and physical function in stroke patients.

## 2. Materials and Methods 

### 2.1. Subjects

The eligible patients were 69 persons who were admitted to a ward for rehabilitation during convalescence after a stroke. The inclusion criteria were as follows: cerebral infarction, cerebral hemorrhage, or subarachnoid hemorrhage; age 40 years and older; no deglutition disorder; and able to walk independently or under supervision using an assistive device if needed.

The exclusion criteria were as follows: deglutition disorder; renal disease or diabetes mellitus needing dietary restriction; and cardiac pacemaker, dementia, untreated cardiovascular disorder, depression, or a schizophrenic disorder.

Recruitment was conducted at Showa University Fujigaka Rehabilitation Hospital from 15 April 2019 to 31 March 2020. The follow-up was conducted during the discharge period. All subjects gave their informed consent prior to participating in the study. The study was conducted in accordance with the Declaration of Helsinki, and the protocol was approved by the Showa University Fujigaka Ethics Committee (ID: F2018C74).

### 2.2. Experimental Design

A single-blind randomized experimental design was used, with 2-month periods of supplementation at breakfast or immediately after exercise in the afternoon.

One co-investigator (AK) created the assignment list using computer-generated random numbers in advance. Each participant was allocated a code number in order of recruitment. Randomization was performed using stratified randomization by age group as follows: 40–49 years old, 50–64 years old, and ≥65 years old. The subjects were randomly divided into two groups: the breakfast group (*n* = 23), who ingested a BCAA-rich nutritional supplement at breakfast; and the post-exercise group (*n* = 23), who ingested a BCAA-rich nutritional supplement immediately after exercise in the afternoon. The chief researcher (IK) was informed of the allocation by age group using the number container method by AK. 

### 2.3. Demographic Data

Demographic data were collected from clinical records, including age, sex, body mass index (BMI), diagnosis, Brunnstrom recovery stage (BRS), and Charlson comorbidity index values.

### 2.4. Outcome Measures

Evaluations were conducted before and 2 months after starting supplementation. Investigators were not blind to group allocation. Investigators determined skeletal muscle mass, lower limb isometric strength, grip strength, timed up-and-go test (TUGT), Berg balance scale (BBS), functional independence measure (FIM), energy consumption and intake, number of combined therapy sessions, and nutritional status. 

•Skeletal muscle mass and body fat mass

Measurements of skeletal muscle mass and body fat mass were performed for all patients using the bioelectrical impedance method (InBody S10, InBody Japan, Tokyo, Japan). The participants were measured after 2 min of rest in a supine position, and were instructed not to move or speak during the measurement.

•Muscle strength

The measurement of muscle strength was conducted by a co-investigator (RM) who was independent of the recruitment, intervention, and other data collection.

Lower limb isometric strength measurement was performed on all patients using a muscle training machine (Wel-tonic L series, Minato Medical Science Co., Ltd., Osaka, Japan). Leg press strength was measured while the subject was sitting on the muscle training machine with knees and hips at 90° of flexion. After participants were familiarized with the test procedure, 2 trials at maximum effort were performed. The higher value was used for analysis.

He grip strength of all patients was measured on the dominant side using a Smedley type grip dynamometer (Grip-D, Takei Scientific Instruments Co., Ltd., Niigata, Japan). Two trials at maximum effort were performed, and the higher value was used for analysis.

•Balance ability

Dynamic balance ability was measured by the BBS and TUGT. Each test was done twice, and the higher value of the BBS and lower value of the TUGT were used. The BBS was recommended in combination with another balance scale [16].

•Energy consumption and intake

Energy consumption was measured using a triaxial accelerometer (Active style PRO HJA-750C, OMRON Corporation, Kyoto, Japan). The accelerometer was attached on the patient’s trunk for 5 days, measurements over 24 h for 3 days were used (excluding the dates of attachment and collection), and the median value was used for analysis.

Energy intake was evaluated by the nurse after each meal and divided into a staple food and a side meal. The intake ratio was recorded as a percentage in the nursing records. Energy intake per day was calculated from the prescribed energy amount and the intake ratio of staple foods and side dishes. Participants were provided nutritional assessment and advice by a dietitian. This study was conducted without restricting the intake of food and drink other than those provided by the hospital.

The energy sufficient ratio was calculated by the quotient of energy consumption and intake.

### 2.5. Interventions

•BCAA supplementation

A leucine-enriched nutritional supplement was provided every day for participants in the breakfast and post-exercise groups. Both groups ingested 125 mL of supplement (200 kcal; Hepas, CLINICO Co., Ltd., Tokyo, Japan). The supplement contained 3.5 g of amino acids and 6.5 g of protein, along with 40 IU of vitamin D per 125 mL. The ratio of amino acids was as follows: 1.6 g leucine, 0.9 g isoleucine, 1.1 g valine; the content percentage of leucine was 44.8%.

Supplementation in the breakfast group was provided at 08:00 with breakfast, and in the post-exercise group just after the exercise therapy session in the afternoon at 14:00–18:00.

•Exercise

The exercise intervention was performed in the rehabilitation hospital, with 2 sessions every day for 2 months for both groups. The rehabilitation sessions contained a 2-session exercise menu: 1 session of physical therapy (one 20-min set each of muscle conditioning exercises, range of motion exercises, and gait and activities of daily living training) and 1 session of occupational therapy (one 20-min set each of range of motion and muscle conditioning exercise, activities using the upper limb, and activities of daily living training).

### 2.6. Statistical Analysis

Statistical analysis was conducted by a co-investigator (AK) who was independent of the recruitment, intervention, and data collection. Based on a study by Yoshimura [10], the minimal sample size was calculated using two-way repeated measures analysis of variance to examine significant differences between the groups (α = 0.05, power = 0.8, effect size = 0.5); 23 participants per group were required. The two groups were created by random assignment of the time of supplementation, including a breakfast group (*n* = 23) and post-exercise group (*n* = 23).

An intention-to-treat analysis was conducted for the groups. The data for participants who dropped out of the intervention were replaced by the “last observation carried forward” method.

An unpaired *t*-test was used to determine the significance of differences between the groups in age, BMI, comorbidity index, energy efficiency ratio, FIM score, number of exercise and nutritional interventions, and nutritional status. The chi-squared test was used to determine the significance of differences between the groups in sex, diagnosis, and BRS. 

Skeletal muscle mass, body fat mass, leg press strength, grip strength, BBS, TUGT, and FIM scores were analyzed by two-way repeated-measures ANOVA (group × time). The interaction was evaluated by time of supplementation and exercise therapy. Items for which a main effect in a group was observed were compared between the groups using the unpaired *t*-test. All data were analyzed using JMP software (version 15, SAS Institute Japan Ltd., Minato, Tokyo, Japan).

## 3. Results

A total of 46 persons participated in this study; 23 persons were excluded, since 20 met the exclusion criteria, and 3 declined participation. Six participants were unable to complete the study after randomization (Figure 1), and no participants had adverse events associated with BCAA supplementation. The demographic data for the participants were similar between the two groups (Table 1 and Table 2), and 31% of patients had obesity (BMI ≥ 25 kg/m^2^). 

There were significant or marginally significant differences in the main effects between the groups in body fat mass (time of supplementation: p = 0.02; confidence interval (CI): 13.2–17.7; average (avg.): 15.5 kg), leg press strength (pre and post: p = 0.03; time of supplementation: p = 0.04; CI: 94.5–124.6; avg.: 109.6 kgf), and BBS (pre and post: p = 0.07; time of supplementation: p = 0.03; CI: 41.6–52.6; avg.: 47.2 points) (Table 3). There were no significant interactions between the groups.

The amount of change from baseline body fat mass (breakfast group: –2.5 ± 2.6 kg; post-exercise group: –0.9 ± 2.1 kg; p = 0.03) was significantly greater in the breakfast group (Table 4), and the change in leg press strength (breakfast group: 26.7 ± 28.5 kgf; post-exercise group: 12.8 ± 18.1 kgf; p = 0.05) was marginally significantly higher in the breakfast group (Table 4). The others did not show significant differences between the two groups (Table 4).

## 4. Discussion

Recently, combined intervention has been viewed with keen interest; the effects of combined intervention on improving muscle strength or mass has gained a certain consensus [1,2,3,4,5,6,17,18], and the range of its use is expanding. Previous studies were done in patients with stroke [10], hip fracture [19], and artificial joint replacement [20,21].

In the present study, whether the time of BCAA ingestion was at breakfast or post-exercise, there was no significant main effect or interaction with skeletal muscle mass. This result is in agreement with multiple previous studies [2,14,18,19,20,21,22,23,24]. Abe et al. [25] reported that stroke patients had decreased skeletal muscle mass on both the affected and unaffected sides of the body starting one month after onset. Although skeletal muscle mass in the present study did not change significantly before or after the combined intervention, it seems that BCAA supplementation prevented a decrease of skeletal muscle mass in both groups. Therefore, the combined intervention for the purpose of increasing muscle strength in stroke patients appeared to be equally effective with breakfast or immediately after exercise in the afternoon.

In the area of applied physiology, researchers have begun to find that BCAA or protein supplementation does not need to be given immediately after exercise, which was previously considered the gold standard [8,13,22,23,24,26,27]. Regarding muscle protein synthesis, the effectiveness of supplementation has been reported when given at breakfast [14], pre-exercise [24], at dinner [26], or before sleep [27]. Ikeda [2] has shown the feasibility of pre-exercise supplementation for muscle strengthening in frail older persons.

In these studies, it was common for supplementation to be combined with exercise intervention, although the time of supplementation varied (breakfast, pre-exercise, post-exercise, or pre-sleep). Atherton et al. reported the “muscle-full” phenomenon [28], in which muscle protein synthesis peaked despite sustained elevation of serum amino acids, and exercise combined with amino acid ingestion caused an increase in amino acid uptake response and sensitization of skeletal muscle to amino acids for up to 24 h [29].

On the other hand, there were significant main effects on lower limb strength and balance ability according to the time of supplementation, which suggests the effectiveness of ingesting BCAA with breakfast for physical ability. In addition, recent research on metabolic syndrome has reported that protein intake before breakfast promotes post-prandial carbohydrate and lipid metabolism [15]. In the present study, 31% of patients had obesity (BMI ≥ 25 kg/m^2^), while body fat mass in the breakfast group was significantly decreased. In Japanese patients, obesity is one of the risk factors for stroke [30]. Considering the prevention of stroke recurrence, if the timing of supplementation does not affect muscle gain, it may be desirable to take BCAAs with breakfast, which is expected to reduce body fat mass.

This study has several limitations, including: (1) there was no control group only taking BCAA without exercise or doing exercise training; (2) some participants dropped out during the follow-up; and (3) for some patients, energy intake was controlled because of diabetes mellitus or obesity. 

First, the design of the present study is lacking in that it does not allow us to separate the effects of nutritional supplementation and exercise, because there was no control group only taking BCAA without exercise or doing exercise training. It has not been possible to examine the effects on body composition and physical function when consuming BCAAs alone, because it is ethically questionable to not provide rehabilitation. The same issue was seen in previous studies [15,16]. Regarding frail [3] and non-frail [31] older persons, the recent systematic reviews showed that protein supplementation alone did not improve muscle mass, muscle strength, or physical function. The same may occur in post-stroke patients.

Second, the data of participants who dropped out of the intervention were replaced by the last observation carried forward method. Even if v intention to treat (ITT) analysis was performed in this study, there might be some biases regarding sample size.

Third, in regard to restricted energy intake, the energy sufficient ratio was approximately 90% in the present study. However, there were no cases of malnutrition in the groups. Therefore, the effect of energy deficiency was relatively small.

## 5. Conclusions

Although the effect of leucine-enriched BCAA supplementation on muscle mass was similar in both groups, a combination of BCAA intake with breakfast and an exercise program was effective at improving physical performance and decreasing body fat mass. The results suggest that ingestion of BCAAs with breakfast is effective for promoting rehabilitation of post-stroke patients. Further study is needed to investigate the ideal timing of BCAA supplementation for stroke. 

## Figures and Tables

**Figure 1 nutrients-12-01928-f001:**
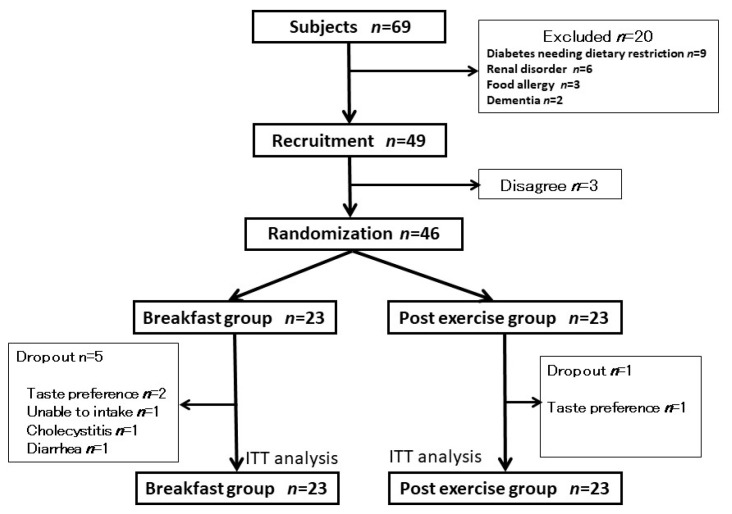
Flowchart of participants in randomized controlled trial of the timing of branched chain amino acid (BCAA) and vitamin D supplementation with exercise therapy. ITT, intention to treat.

**Table 1 nutrients-12-01928-t001:** Baseline demographic data for the participants.

	Breakfast Group	Post-Exercise Group	*p*-Value
	(*n* = 23)	(*n* = 23)	
Age (years)	65.5 ± 13.1	67.5 ± 5	0.635
Sex (male/female)	14:09	14:09	1.0
Body mass index (kg/m^2^)	23.2 ± 3.3	21.3 ± 4.3	0.093
Charlson co-morbidity index (score)	4.7 ± 1.7	5.0 ± 1.9	0.464
Diagnosis: Cerebral infarction	14 persons	13 persons	
Cerebral hemorrhage	5 persons	7 persons	0.773
Subarachnoid hemorrhage	4 persons	3 persons	
Brumstrom recovery stage (I:II:III:IV:V:VI)			
Upper limb	0:3:3:3:5:9	0:2:5:3:3:10	0.860
Lower limb	0:3:3:3:4:11	0:4:5:3:4:7	0.772
Finger	1:0:4:4:5:9	0:2:6:4:2:9	0.455

Note: Mean ± standard deviation. There were no significant differences between groups.

**Table 2 nutrients-12-01928-t002:** Comparison of number of sessions, nutritional status, and energy status between the two groups.

	Breakfast Group	Post-Exercise Group	*p*-Value
	(*n* = 23)	(*n* = 23)	
Number of sessions Exercise therapy	89.0 ± 34.8	87.7 ± 32.8	0.900
Supplementation	42.3 ± 19.6	43.0 ± 16.3	0.893
Nutritional status			
Albumin (g/dL): Pre-intervention	3.8 ± 0.6	3.8 ± 0.6	0.980
Post-intervention	4.0 ± 0.4	3.9 ± 0.7	0.578
Total protein (g/dL): Pre-intervention	6.8 ± 0.6	6.7 ± 0.7	0.456
Post-intervention	7.0 ± 0.5	6.9 ± 0.6	0.319
Energy status at baseline (kcal/day)			
Energy consumption	1865.8 ± 422.5	1767.9 ± 475.1	0.464
Energy intake	1611.5 ± 331.6	1581.6 ± 278.5	0.742
Energy sufficient ratio (%)	88.3 ± 18.1	93.6± 23.1	0.399

Note: Mean ± standard deviation. There were no significant differences between groups.

**Table 3 nutrients-12-01928-t003:** Group × time analysis of body composition, muscle strength, physical function, and functional independence measure.

	Group	Pre-Intervention	Post-Intervention	Main Effect (Group)	95% CI	
				*p*-Value	Lower Limit	Upper Limit
Skeletal muscle mass (kg)	Breakfast	23.7 ± 5.0	23.9 ± 4.6	0.679	21.5	26.1
	Post-exercise	23.4 ± 7.4	23.1 ± 7.8			
Body fat mass (kg)	Breakfast	17.6 ± 6.6	15.1 ± 6.4	0.023 *	13.2	17.7
	Post-exercise	13.9 ± 6.6	12.9 ± 5.5			
Leg press strength (kgf)	Breakfast	86.3 ± 45.9	113.0 ± 35.4	0.038 *	94.5	124.6
	Post-exercise	74.9 ± 44.1	87.7 ± 41.7			
Grip strength (kgf)	Breakfast	25.7 ± 9.9	26.2 ± 8.7	0.290	22.6	29.9
	Post-exercise	23.4 ± 11.3	24.0 ± 10.8			
Berg balance scale (score)	Breakfast	41.3 ± 15.4	47.2 ± 8.4	0.030 *	40.8	53.6
	Post-exercise	34.3 ± 19.4	40.0 ± 16.2			
Timed up and go test (s)	Breakfast	18.8 ± 19.6	19.0 ± 19.1	0.081 ^†^	2.9	33.1
	Post-exercise	36.2 ± 56.7	32.5 ± 56.3			
Functional independence	Breakfast	62.9 ± 18.9	80.6 ± 10.8	0.072 ^†^	72.6	85.4
Measure (score)	Post-exercise	59.3 ± 19.7	70.7 ± 20.1			

Note: CI, confidence interval. Mean ± standard deviation. * *p* < 0.05; † *p* < 0.1.

**Table 4 nutrients-12-01928-t004:** Comparison of amount of change from baseline between the two groups.

	Breakfast Group (*n* = 23)	Post-Exercise Group (*n* = 23)	*p*-Value
Δ Body fat mass (kg)	–2.5 ± 2.6	–0.9 ± 2.1	0.038 *
Δ Leg press strength (kgf)	26.7 ± 28.5	12.8 ± 18.1	0.051 ^†^
Δ Berg balance scale (score)	5.9 ± 10.2	5.8 ± 9.8	0.96

Note: Mean ± standard deviation. * *p* < 0.05; ^†^
*p* < 0.1.

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
