# Peer review of "The Effects of Timing of a Leucine-Enriched Amino Acid Supplement on Body Composition and Physical Function in Stroke Patients: A Randomized Controlled Trial"

_nutrients, 2020, doi:10.3390/nu12071928_

Round 1

Reviewer 1 Report

In this original research manuscript, the authors described the effect of the timing of supplemental BCAAs with exercise therapy on physical function in stroke patients. Overall, it is an interesting study but some issues need to be addressed for full consideration. I suggest an English editing. 

General comments:

Table 2, which is mentioned in the paper, is missing. The authors switched from table 1 to table 3.

Specific comments:

Page 1; Line 23-24: This sentence needs rephrasing. Additionally, in my opinion, the term “elderly” should be avoided since it is pejorative and reductionist. I suggest another term (i.e., older people, older adults, older persons…)

Page 2; Line 42: Change the term “elderly” to something else

Page 2; Line 46: Change the term “elderly” to something else

Page 2; Line 44-68: I suggest to include the findings of a recent systematic review and meta-analysis which explored the associations between muscle mass gain and protein supplementation and muscle strengthening exercise efficacy in older people with a high risk of sarcopenia or frailty (Liao CD et al. Nutrients 2019, 11, 1713; doi:10.3390/nu11081713). I suggest also to include something about a recent article regarding nutrition and physical exercise interventions in older people to counteract sarcopenia (Damanti S. et al. Nutrients 2019 Aug 23;11(9):1991.  doi: 10.3390/nu11091991.).

Page 2; Line 79-83: Please rephrase. In particular, the procedure of the informed consent is repeated twice.

Page 5; Line 140-144: It is mentioned that the evaluation of energy intake was carried out. However, along the paper there are no data about this aspect and the method used to assess energy intake (i.e., 24h-recall or food record) is not explained. I suggest including also if participants received nutritional advices. It would be interesting to know the intake of nutrient and protein with the diet (if you have it).

Page 6; Line 151-153: Did the supplement contain both 6500 mg of proteins and 3500 mg of BCAA? Or 6500 mg of proteins of which 3500 mg of BCAA? Furthermore, I suggest changing units from mg to g.

Page 7; Line 204: I suggest changing “therapy” to “intervention”

Page 7; Line 208-210: I suggest you take a look also to the two consensus papers from the PROT-AGE study group (Bauer J et al. J Am Med Dir Assoc. 2013 Aug;14(8):542-59. doi: 10.1016/j.jamda.2013.05.021.) and from the European Society for Clinical Nutrition and Metabolism (Deutz et al. Clin Nutr . 2014 Dec;33(6):929-36. doi: 10.1016/j.clnu.2014.04.007.)

Page 7; Line 210-219: I suggest adding something about the protein type and quantity to better stimulate muscle accretion. (Bauer J et al. J Am Med Dir Assoc. 2013 Aug;14(8):542-59. doi: 10.1016/j.jamda.2013.05.021.) (Damanti S. et al. Nutrients 2019 Aug 23;11(9):1991.  doi: 10.3390/nu11091991)

Page 8; Line 226-228: This sentence needs rephrasing as does not flow well.

Page 8; Line 233-234: Please rephrase

Page 8; Line 235-236: “the participants in this study also had a 31% of patients with obesity” Please rephrase.

Page 8; Line 243: Change the term “elderly” to something else

Page 8; Line 246: Change the term “elderly” to something else

Page 9; Line 255-257: Please rephrase as does not flow well.

Author Response

We thank the referees for their helpful suggestions. We have revised the manuscript on the basis of the reviewers’ comments.

This manuscript has checked by English pre-edit services on MDPI because of the suggestion to need editing by a native English speaker.

Our responses to the reviewers’ comments are as follows:

Reviewer 1

Comment 1)

  Table 2, which is mentioned in the paper, is missing. The authors switched from table 1 to table 3.

Response to comment 1)

We had forgot to attach Table 2. We have added Table 2 to “Results” (page 6, lines 196-198).

Comment 2)

  Page 1; Line 23-24: This sentence needs rephrasing. Additionally, in my opinion, the term “elderly” should be avoided since it is pejorative and reductionist. I suggest another term (i.e., older people, older adults, older persons…)

Response to comment 2)

Thank you very much for this comment. The sentence “elderly” have rephrased to “older person” in this manuscript, please confirm.

Comment 3)

Page 2; Line 44-68: I suggest to include the findings of a recent systematic review and meta-analysis which explored the associations between muscle mass gain and protein supplementation and muscle strengthening exercise efficacy in older people with a high risk of sarcopenia or frailty (Liao CD et al. Nutrients 2019, 11, 1713; doi:10.3390/nu11081713). I suggest also to include something about a recent article regarding nutrition and physical exercise interventions in older people to counteract sarcopenia (Damanti S. et al. Nutrients 2019 Aug 23;11(9):1991.  doi: 10.3390/nu11091991.).

Response to comment 3)

Thank you for the suggestion regarding references. We have added these papers to “Introduction” (page 2, lines 42-48) and “References” No.7,8.

Comment 4)

Page 2; Line 79-83: Please rephrase. In particular, the procedure of the informed consent is repeated twice.

Response to comment 4)

“The procedure of the informed consent” was corrected (page2-3, lines 78-82). And, this manuscript has checked by English pre-edit services on MDPI.

Comment 5)

Page 5; Line 140-144: It is mentioned that the evaluation of energy intake was carried out. However, along the paper there are no data about this aspect and the method used to assess energy intake (i.e., 24h-recall or food record) is not explained. I suggest including also if participants received nutritional advices. It would be interesting to know the intake of nutrient and protein with the diet (if you have it).

Response to comment 5)

Thank you for the suggestion. We have added the information “the method used to assess energy intake” (page 4, lines 135-136) and “nutritional advices.” (page 4, line 138).

Comment 6)

Page 6; Line 151-153: Did the supplement contain both 6500 mg of proteins and 3500 mg of BCAA? Or 6500 mg of proteins of which 3500 mg of BCAA? Furthermore, I suggest changing units from mg to g.

Response to comment 6)

We have corrected this sentence and changed the units from “mg” to “g” (page4, lines 147-149).

Comment 7)

Page 7; Line 204: I suggest changing “therapy” to “intervention”

Response to comment 7)

Thank you for this comment. The sentence “therapy” have rephrased to “intervention” in this manuscript, please confirm.

Comment 8)

Page 7; Line 208-210: I suggest you take a look also to the two consensus papers from the PROT-AGE study group (Bauer J et al. J Am Med Dir Assoc. 2013 Aug;14(8):542-59. doi: 10.1016/j.jamda.2013.05.021.) and from the European Society for Clinical Nutrition and Metabolism (Deutz et al. Clin Nutr . 2014 Dec;33(6):929-36. doi: 10.1016/j.clnu.2014.04.007.)

Page 7; Line 210-219: I suggest adding something about the protein type and quantity to better stimulate muscle accretion. (Bauer J et al. J Am Med Dir Assoc. 2013 Aug;14(8):542-59. doi: 10.1016/j.jamda.2013.05.021.) (Damanti S. et al. Nutrients 2019 Aug 23;11(9):1991.  doi: 10.3390/nu11091991)

Response to comment 8)

Thank you for the suggestion regarding references. We have added these papers to “Discussion” (page 8, line 212 and 226) and “References” No.8,17,18.

Comment 9)

Page 8; Line 235-236: “the participants in this study also had a 31% of patients with obesity” Please rephrase.

Response to comment 9)

The information was added to “Results” and, this manuscript has checked by English pre-edit services on MDPI.

Comment 10)

Page 8; Line 226-228: This sentence needs rephrasing as does not flow well.

Page 8; Line 233-234: Please rephrase

Page 9; Line 255-257: Please rephrase as does not flow well.

Response to comment 10)

  This manuscript has checked by English pre-edit services on MDPI because of the suggestion to need editing by a native English speaker.

Reviewer 2 Report

Thank you for the possibility to review the article: “The effect of timing of a leucine-enriched amino acid supplement on body composition and physical function in stroke patients”

The topic of this study is of great interest. To date, little is known about the efficacy of nutritional support on body composition and physical function in patients during post-stroke rehabilitation. This randomized controlled trial aimed to evaluate the effect of the timing (with breakfast or post-exercise) of BCAA supplementation on body composition and physical function in post-stroke patients.

However, the manuscript appears confusing and needs to be re-worked in terms of both form and substance. Here you can find a list of comments and major revisions.

1) Title

L4: Please revise it “in inpatient stroke patients”

2) The abstract is confusing and needs to be reviewed and clarified.

L24: Previously – as previously shown?

L28: Please reformulate “on when supplementation”

L27-32: the period is unclear, please reformulate

L32-33: please specify the p-value and confidence interval.

3) Introduction

L50, 51, 53 … : when you cite an author, it would be advisable to add “et al.” after the last name.

L60 and L66-67: vitamin D? Vitamin D appears out of context. If you choose to evaluate the effect of timing of vitamin D supplementation, it would be advisable to introduce it at the beginning of the introduction and discuss it in the results and discussion.

L68:  in patients stroke patients?

The introduction needs to be revised.

4) Materials and methods

Substantially, this study has several biases such as small sample size (with important drop-out during the follow-up, even if ITT analysis was performed). Furthermore, a control group without exercise training or a control group only with exercise training is lacking not allowing to separating the effects of timing of nutritional supplementation and exercise. Moreover, there is some confusion between method and results parts, and some outcomes are not reported in the results such as nutritional status or energy intake.

L72: the inclusion criteria need to be clarified.

L76 -77:  the number of excluded and included patients should be specified in the results part.

Table 1: the baseline demographic data should be placed in the results part.

Table 1: “no differences between groups”, did you mean “no significant” differences? please specify the p-value.

Figure 1:  The flowchart should be placed in the results part.

L107-109:  The collected clinical records should not be placed in the paragraph outcomes measures.

L128: which arm was evaluated for the grip test?

Results:

L185: Table 2 as you indicated is missing.

Energy consumption and intake outcomes as indicated as L106 are lacking in the results.

Nutritional status outcomes as indicated by L 107 are lacking in the results.

The outcome “Number of combined therapy sessions” as indicated by L107 were not presented in the results.

Discussion:

L220-228: This paragraph presents the results and should be placed at the beginning of the discussion.

L239-249: this paragraph presenting a mix of different limitations needs to be reviewed and re-organized to clarify the reading.

L246-249: these results have never been introduced in the manuscript, before. Please introduce them to the results paragraph.

Moreover, to make easier the reading and understanding, the paper should be carefully revised by a native English speaker.

Conclusions:

L254: Please add “with exercise program” since a control without exercise training or a control group only with exercise training is lacking not allowing to separating the effects of nutritional supplementation and exercise.

L256: you evaluated stroke patients, mentioning “sarcopenic or frail persons, and hip fracture patients “are out of context.

Author Response

We thank the referees for their helpful suggestions.

We have revised the manuscript on the basis of the reviewers’ comments.

This manuscript has checked by English pre-edit services on MDPI because of the suggestion to need editing by a native English speaker.

Our responses to the reviewers’ comments are as follows:

Reviewer 2

Comment 1)

1) Title L4: Please revise it “in inpatient stroke patients”

Response to comment 1)

  We have corrected the Title to “The effect of timing of a leucine-enriched amino acid supplement on body composition and physical function in stroke patients: A randomized controlled trial”

Comment 2)

  2) The abstract is confusing and needs to be reviewed and clarified.

L24: Previously – as previously shown?      L28: Please reformulate “on when supplementation”

L27-32: the period is unclear, please reformulate

Response to comment 2)

  Thank you for your suggestion. We have reformulated the abstract regarding these points.

  This manuscript has checked by English pre-edit services on MDPI because of the suggestion to need editing by a native English speaker.

L32-33: please specify the p-value and confidence interval.

Response to comment 2)

  Thank you for the comment. We have added the information of p-value and CI to “Abstract” and “Results (Table 3)”.

Comment 3)

3) Introduction L50, 51, 53 … : when you cite an author, it would be advisable to add “et al.” after the last name.

Response to comment 3)

  We have corrected as instructed regarding citation in this manuscript.

Comment 4)

  L60 and L66-67: vitamin D? Vitamin D appears out of context. If you choose to evaluate the effect of timing of vitamin D supplementation, it would be advisable to introduce it at the beginning of the introduction and discuss it in the results and discussion.

  L68:  in patients stroke patients?

Response to comment 4)

  Thank you for bringing this to our attention. We have corrected this point as follows:

  L60: a recent randomized study [14] reported that protein supplementation with breakfast stimulated postprandial muscle protein synthesis and increased muscle mass in healthy older persons.

L66-67: This study therefore investigated the effect of the timing of BCAA supplementation with exercise therapy on improving skeletal muscle mass, muscle strength, and physical function in stroke patients.

Comment 5)

4) Materials and methods: Substantially, this study has several biases such as small sample size (with important drop-out during the follow-up, even if ITT analysis was performed). Furthermore, a control group without exercise training or a control group only with exercise training is lacking not allowing to separating the effects of timing of nutritional supplementation and exercise.

L239-249: this paragraph presenting a mix of different limitations needs to be reviewed and re-organized to clarify the reading.

Response to comment 5)

Thank you for the suggestion. On the basis of the comment 4), We have added the information of these biases to “Discussion” limitation 1) and 2). (Page 8-9, lines 245-258). And “limitation” re-organized to clarify the reading.

Comment 6)

L72: the inclusion criteria need to be clarified.

Response to comment 6)

  The inclusion criteria are clarified as follows: Page 2, lines 72-74.

  The inclusion criteria were as follows: cerebral infarction, cerebral hemorrhage, or subarachnoid hemorrhage; age 40 years and older; no deglutition disorder; and able to walk independently or under supervision using an assistive device if needed.

Comment 7)

L76 -77:  the number of excluded and included patients should be specified in the results part.

Table 1: the baseline demographic data should be placed in the results part

Figure 1:  The flowchart should be placed in the results part.

L107-109:  The collected clinical records should not be placed in the paragraph outcomes measures.

Response to comment 7)

  Thank you for the suggestions. On the basis of the comment, these points have replaced to the results part.

Comment 8)

Table 1: “no differences between groups”, did you mean “no significant” differences? please specify the p-value.

Response to comment 8)

  The legend was corrected to “no significant differences between groups”.

  We have added the information of the p-value on Table 1-3.

Comment 9)

  L128: which arm was evaluated for the grip test?

Response to comment 9)

  Thank you for the comment. This point is corrected as follow:

  Grip strength of all patients was measured on the dominant side using a Smedley type grip dynamometer

  (Page 4, lines 121-122)

Comment 10)

L185: Table 2 as you indicated is missing.

Energy consumption and intake outcomes as indicated as L106 are lacking in the results.

Nutritional status outcomes as indicated by L 107 are lacking in the results.

The outcome “Number of combined therapy sessions” as indicated by L107 were not presented in the results.

Response to comment 10)

We had forgot to attach Table 2. We have added Table 2 to “Results” (page 6, lines 196-198). These outcomes are containing on Table 2.

Comment 11)

L220-228: This paragraph presents the results and should be placed at the beginning of the discussion.

Response to comment 11)

  Thank you for the suggestion. The paragraph is replaced to the beginning of the discussion.

Comment 12)

L246-249: these results have never been introduced in the manuscript, before. Please introduce them to the results paragraph.

Response to comment 12)

  The information was added to “Results” (Page 5, line 185)

Comment 13)

Moreover, to make easier the reading and understanding, the paper should be carefully revised by a native English speaker.

Response to comment 13)

  This manuscript has checked by English pre-edit services on MDPI because of the suggestion to need editing by a native English speaker.

Comment 14)

  L254: Please add “with exercise program” since a control without exercise training or a control group only with exercise training is lacking not allowing to separating the effects of nutritional supplementation and exercise.

  L256: you evaluated stroke patients, mentioning “sarcopenic or frail persons, and hip fracture patients “are out of context.

Response to comment 14)

  Thank you for this suggestion. We have corrected “Conclusion” as follows:

  Although the effect of leucine-enriched BCAA supplementation on muscle mass was similar in both groups, a combination of BCAA intake with breakfast and an exercise program was effective at improving physical performance and decreasing body fat mass. The results suggest that ingestion of BCAAs with breakfast is effective for promoting rehabilitation of post-stroke patients. Further study is needed to investigate the ideal timing of BCAA supplementation for stroke.

Round 2

Reviewer 1 Report

all my comments have been addressed

Reviewer 2 Report

The manuscript has now improved significantly.